# Redefining Temporal Modeling in Video Diffusion: The Vectorized Timestep Approach

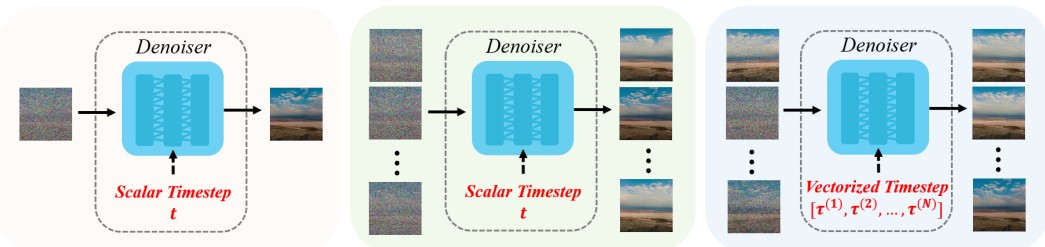

|     |     |     |
| --- | --- | --- |
| (a) Image Diffusion Model | (b) Conventional Video Diffusion Model | (c) Proposed FVDM |

Figure 1: Previous conventional video diffusion models (b) directly extend image diffusion models (a) utilizing a single scalar timestep on the whole video clip. This straightforward adaption restricts the flexibilities of VDM's in downstream tasks, *e.g.*, image-to-video generation, longer video generation. In this paper, we propose Frame-Aware Video Diffusion Model (FVDM), which trains the denoiser via a vectorized timestep variable (c). Our method attains superior visual quality not only in standard video generation but also enables multiple downstream tasks in a zero-shot manner.

## Abstract

Diffusion models have revolutionized image generation, and their extension to video generation has shown promise. However, current video diffusion models (VDMs) rely on a scalar timestep variable applied at the clip level, which limits their ability to model complex temporal dependencies needed for various tasks like image-to-video generation. To address this limitation, we propose a frame-aware video diffusion model (FVDM), which introduces a novel vectorized timestep variable (VTV). Unlike conventional VDMs, our approach allows each frame to follow an independent noise schedule, enhancing the model's capacity to capture fine-grained temporal dependencies. FVDM's flexibility is demonstrated across multiple tasks, including standard video generation, image-to-video generation, video interpolation, and long video synthesis. Through a diverse set of VTV configurations, we achieve superior quality in generated videos, overcoming challenges such as catastrophic forgetting during fine-tuning and limited generalizability in zero-shot methods. Our empirical evaluations show that FVDM outperforms state-of-the-art methods in video generation quality, while also excelling in extended tasks. By addressing fundamental shortcomings in existing VDMs, FVDM sets a new paradigm in video synthesis, offering a robust framework with significant implications for generative modeling and multimedia applications.

## 1 Introduction

The advent of diffusion models (Song et al., 2020b; Ho et al., 2020) has heralded a paradigm shift in generative modeling, particularly in the domain of image synthesis. These models, which leverage an iterative noise reduction process, have demonstrated remarkable efficacy in producing high-fidelity samples. Naturally, we can extend this framework to video generation (Ho et al., 2022; He et al., 2022; Chen et al., 2023a; Wang et al., 2023; Ma et al., 2024; OpenAI, 2024; Xing et al., 2023b) by denoising a whole video clip jointly. These methods have shown promising results, yet it has also exposed fundamental limitations in modeling the complex temporal dynamics inherent to video data.

The crux of the problem lies in the naive adaptation of image diffusion principles to the video domain. As shown in Fig. 1, conventional video diffusion models (VDMs) typically treat a video as a monolithic entity, employing a scalar timestep variable to govern the diffusion process uniformly across all frames following image diffusion models. While this approach has proven adequate for generating short video clips, it fails to capture the nuanced temporal dependencies that characterize real-world video sequences. This limitation not only constrains the model's flexibility but also impedes its scalability in handling more sophisticated temporal structures.

The temporal modeling deficiency of current VDMs has spawned a plethora of task-specific adaptations, particularly in domains such as image-to-video generation (Xing et al., 2023a; Guo et al., 2023; Ni et al., 2024), video interpolation (Wang et al., 2024a;b), and long video generation (Qiu et al., 2023; Henschel et al., 2024). These approaches have largely relied on two primary strategies: fine-tuning and zero-shot techniques. For instance, DynamiCrafter (Xing et al., 2023a) achieves open-domain image animation through fine-tuning a pre-trained VDM (Chen et al., 2023a) conditioned on input images. In the realm of video interpolation, Wang et al. (2024b) propose a lightweight fine-tuning technique coupled with a bidirectional diffusion sampling process. Concurrently, zero-shot methods such as DDIM inversion (Mokady et al., 2023) and noise rescheduling (Qiu et al., 2023) have been employed to adapt pretrained VDMs for tasks like image-to-video generation (Ni et al., 2024) and long video synthesis (Qiu et al., 2023). However, these approaches often grapple with issues such as catastrophic forgetting during fine-tuning or limited generalizability in zero-shot scenarios, resulting in suboptimal utilization of the VDMs' latent capabilities.

To address these fundamental limitations, we introduce a novel framework: the *frame-aware video diffusion model* (FVDM). At the heart of our approach lies a *vectorized timestep variable* (VTV) that enables independent frame evolution (shown in Fig. 1(c)). This stands in stark contrast to existing VDMs, which rely on a scalar timestep variable that enforces uniform temporal dynamics across all frames. Our innovation allows each frame to traverse its own temporal trajectory during the forward process while simultaneously recovering from noise to the complete video sequence in the reverse process. This paradigm shift significantly enhances the model's capacity to capture intricate temporal dependencies and markedly improves the quality of generated videos.

The contributions of our work are threefold:

**Enhanced Temporal Modeling:** Introducing the Frame-Aware Video Diffusion Model (FVDM), which utilizes a vectorized timestep variable (VTV) to enable independent frame evolution and superior temporal dependency modeling.

**Numerous (Zero-Shot) Applications:** FVDM's flexible VTV configurations support a wide array of tasks, including standard video synthesis (i.e., synthesizing video clips), image-to-video transitions, video interpolation, long video generation, and so on, all without re-training.

**Superior Performance Validation:** Our empirical evaluations demonstrate that FVDM not only exceeds current state-of-the-art methods in video quality for standard video generation but also excels in various extended applications, highlighting its robustness and versatility.

Our proposed FVDM represents a significant advancement in the field of video generation, offering a powerful and flexible framework that opens new avenues for both theoretical exploration and practical application in generative modeling. By addressing the fundamental limitations of existing VDMs, FVDM paves the way for more sophisticated and temporally coherent video synthesis, with far-reaching implications for various domains in computer vision and multimedia processing.

## 2 METHODS

### 2.1 PRELIMINARIES: DIFFUSION MODELS

Diffusion models have emerged as a powerful framework for generative modeling, grounded in the theory of stochastic differential equations (SDEs). These models generate data by progressively adding noise to the data distribution and then reversing this process to sample from the noise distribution (Song et al., 2020b; Karras et al., 2022). In the following, we provide a foundational understanding of diffusion models, essential to our work.

At the core of diffusion models is the concept of data diffusion, where the original data distribution $p_{\text{data}}(\mathbf{x})$ is perturbed over time $t \in [0, T]$ via a continuous process governed by an SDE:

$$d\mathbf{x} = \boldsymbol{\mu}(\mathbf{x}, t)\, dt + \sigma(t)\, d\mathbf{w}, \tag{1}$$

where $\boldsymbol{\mu}(\cdot, \cdot)$ and $\sigma(\cdot)$ represent the drift and diffusion coefficients, and $\{\mathbf{w}(t)\}_{t \in [0, T]}$ denotes the standard Brownian motion. This diffusion process results in a time-dependent distribution $p_t(\mathbf{x}(t))$, with the initial condition $p_0(\mathbf{x}) \equiv p_{\text{data}}(\mathbf{x})$.

The generative process in diffusion models is achieved by reversing the diffusion SDE, allowing sampling from an initially Gaussian noise distribution. This reverse process is characterized by the reverse-time SDE using the score function $\nabla_{\mathbf{x}} \log p_t(\mathbf{x})$:

$$d\mathbf{x} = [\boldsymbol{\mu}(\mathbf{x}, t) - \sigma(t)^2 \nabla_{\mathbf{x}} \log p_t(\mathbf{x})]dt + \sigma(t)d\bar{\mathbf{w}}, \tag{2}$$

where $\bar{\mathbf{w}}$ represents the standard Wiener process in reverse time.

A crucial aspect of this SDE framework is the associated Probability Flow (PF) ODE (Song et al., 2020b), which describes the corresponding deterministic process sharing the same marginal probability densities $\{p_t(\mathbf{x})\}_{t=0}^{T}$ as the SDE:

$$d\mathbf{x} = \left[\boldsymbol{\mu}(\mathbf{x}, t) - \frac{1}{2}\sigma(t)^2 \nabla_{\mathbf{x}} \log p_t(\mathbf{x})\right] dt. \tag{3}$$

In practice, this reverse process involves training a score model to approximate the score function, which is then integrated into the empirical PF ODE for sampling.

While diffusion models have shown promise in various domains, their application to video data presents unique challenges, especially the modeling of high-dimensional temporal data.

## 2.2 Frame-Aware Video Diffusion Model

We present a novel frame-aware video diffusion model that significantly enhances the generative capabilities of traditional diffusion models by introducing a vectorized timestep variable. This approach allows for the independent evolution of each frame in a video clip, capturing complex temporal dependencies and improving performance across various video generation tasks. In this section, we provide a detailed mathematical formulation of our model, its underlying principles, and its applications.

### 2.2.1 Vectorized Timestep Variable

Inherited from image diffusion models, current video diffusion models also employ a scalar time variable $t \in [0, T]$ that applies uniformly across all elements of the data being generated (Xing et al., 2023b). In the context of video generation, this approach fails to capture the nuanced temporal dynamics inherent in video sequences. To address this limitation, we introduce a vectorized timestep variable $\boldsymbol{\tau}(t) : [0, T] \to [0, T]^N$, defined as:

$$\boldsymbol{\tau}(t) = [\tau^{(1)}(t), \tau^{(2)}(t), \dots, \tau^{(N)}(t)]^\top \tag{4}$$

where $N$ is the number of frames in the video sequence, and $\tau^{(i)}(t)$ represents the individual time variable for the $i$-th frame. This vectorization allows for independent noise perturbation for each frame, enabling a more flexible and detailed diffusion process.

### 2.2.2 Forward SDE with Independent Noise Scales

We extend the conventional forward stochastic differential equation (SDE) to accommodate our vectorized timestep variable. For each frame $\mathbf{x}^{(i)}$, the forward process is governed by:

$$d\mathbf{x}^{(i)} = \boldsymbol{\mu}(\mathbf{x}^{(i)}, \tau^{(i)})dt + \sigma(\tau^{(i)})d\mathbf{w}^{(i)} \tag{5}$$

This formulation allows each frame to experience noise from an independent Gaussian distribution, governed by its specific $\tau^{(i)}(t)$.

For representation simplicity, we integrate all frame SDEs into one single SDE for the whole video. Let's define the video as $\mathbf{X} \in \mathbb{R}^{N \times d}$, where $N$ is the number of frames and $d$ is the dimensionality of each frame. We can represent the video as a matrix:

$$\mathbf{X} = [\mathbf{x}^{(1)}, \mathbf{x}^{(2)}, \dots, \mathbf{x}^{(N)}]^\top \tag{6}$$

where each $\mathbf{x}^{(i)} \in \mathbb{R}^d$ represents a single frame. We can now formulate an integrated forward SDE for the entire video:

$$d\mathbf{X} = \boldsymbol{U}(\mathbf{X}, \boldsymbol{\tau}(t))dt + \boldsymbol{\Sigma}(\boldsymbol{\tau}(t))d\mathbf{W} \tag{7}$$

where $\boldsymbol{U}(\cdot, \boldsymbol{\tau}(\cdot)) : \mathbb{R}^{N \times d} \times [0, T] \to \mathbb{R}^{N \times d}$ is the drift coefficient for the entire video, $\boldsymbol{\Sigma}(\boldsymbol{\tau}(\cdot)) : [0, T] \to \mathbb{R}^{N \times N}$ is a diagonal matrix of diffusion coefficients, $\mathbf{W}$ is an standard Brownian motion.

The drift and diffusion terms can be expressed as:

$$\boldsymbol{U}(\mathbf{X}, \boldsymbol{\tau}(t)) = \left[ \boldsymbol{\mu}(\mathbf{x}^{(1)}, \tau^{(1)}(t)), \boldsymbol{\mu}(\mathbf{x}^{(2)}, \tau^{(2)}(t)), \dots, \boldsymbol{\mu}(\mathbf{x}^{(N)}, \tau^{(N)}(t)) \right]^\top \tag{8}$$

$$\boldsymbol{\Sigma}(\boldsymbol{\tau}(t)) = \begin{bmatrix} \sigma(\tau^{(1)}(t)) & 0 & \cdots & 0 \\ 0 & \sigma(\tau^{(2)}(t)) & \cdots & 0 \\ \vdots & \vdots & \ddots & \vdots \\ 0 & 0 & \cdots & \sigma(\tau^{(N)}(t)) \end{bmatrix} \tag{9}$$

This formulation preserves the independent noise scales for each frame while providing a unified representation for the entire video. In the context of DDPMs (Ho et al., 2020), the drift coefficient $\boldsymbol{\mu}(\mathbf{x}^{(i)}, \tau^{(i)}(t))$ and the diffusion coefficient $\sigma(\tau^{(i)}(t))$ for each frame $i$ (where $1 \le i \le N$) are given by: $\boldsymbol{\mu}(\mathbf{x}^{(i)}, \tau^{(i)}(t)) = -\frac{1}{2}\beta(\tau^{(i)}(t))\mathbf{x}^{(i)}$, $\sigma(\tau^{(i)}(t)) = \sqrt{\beta(\tau^{(i)}(t))}$, where $\beta(\cdot)$ is the noise scale function, which is a predefined non-negative, non-decreasing function that determines the amount of noise added at each timestep $i$ with $\beta(0) = 0.1$ and $\beta(T) = 20$ (Song et al., 2020b).

### 2.2.3 REVERSE SDE AND SCORE FUNCTION

In the context of the reverse process, we define an integrated reverse SDE to encapsulate the dependencies across joint frames:

$$d\mathbf{X} = \left[ \boldsymbol{U}(\mathbf{X}, \boldsymbol{\tau}(t)) - \frac{1}{2}\boldsymbol{\Sigma}(\boldsymbol{\tau}(t))\boldsymbol{\Sigma}(\boldsymbol{\tau}(t))^\top \nabla_{\mathbf{X}} \log p_t(\mathbf{X}) \right] dt + \boldsymbol{\Sigma}(\boldsymbol{\tau}(t))d\bar{\mathbf{W}} \tag{10}$$

where $\bar{\mathbf{W}}$ is an $N$-dimensional standard Brownian motion with $dt < 0$.

The score-based model $\mathbf{s}_\theta(\cdot, \boldsymbol{\tau}(\cdot)) : \mathbb{R}^{N \times d} \times [0, T] \to \mathbb{R}^{N \times d}$ is designed to operate over the entire video sequence. The model's learning objective is to approximate the score function:

$$\mathbf{s}_\theta(\mathbf{X}, \boldsymbol{\tau}(t)) \approx \nabla_{\mathbf{X}} \log p_t(\mathbf{X}) \tag{11}$$

The optimization problem for the model parameters $\theta$ is formulated as:

$$\theta^* = \arg\min_\theta \mathbb{E}_t \mathbb{E}_{\boldsymbol{\tau}(t)} \left[ \lambda(t) \mathbb{E}_{\mathbf{X}(0)} \mathbb{E}_{\mathbf{X}(\boldsymbol{\tau}(t))|\mathbf{X}(0)} \right.$$
$$\left. \left[ \left\| \mathbf{s}_\theta(\mathbf{X}(\boldsymbol{\tau}(t)), \boldsymbol{\tau}(t)) - \nabla_{\mathbf{X}(\boldsymbol{\tau}(t))} \log p_t(\mathbf{X}(\boldsymbol{\tau}(t))|\mathbf{X}(0)) \right\|_2^2 \right] \right] \tag{12}$$

where $\lambda(\cdot)$ is a positive weighting function that can be chosen proportional to $1/\mathbb{E}\left[ \|\nabla_{\mathbf{X}(\boldsymbol{\tau}(t))} \log p_t(\mathbf{X}(\boldsymbol{\tau}(t))|\mathbf{X}(0))\|_2^2 \right]$, as discussed in the context of score matching in Hyvärinen (2005); Särkkä & Solin (2019); Song et al. (2020b).

### 2.3 IMPLEMENTATION

**Network Architecture.** Our proposed method can work for all current VDMs' backbones with small adaptation. For the sake of simplicity, we choose a novel video diffusion transformer model developed by Ma et al. (2024) as our backbone in this work. To adapt the scalar timestep variable

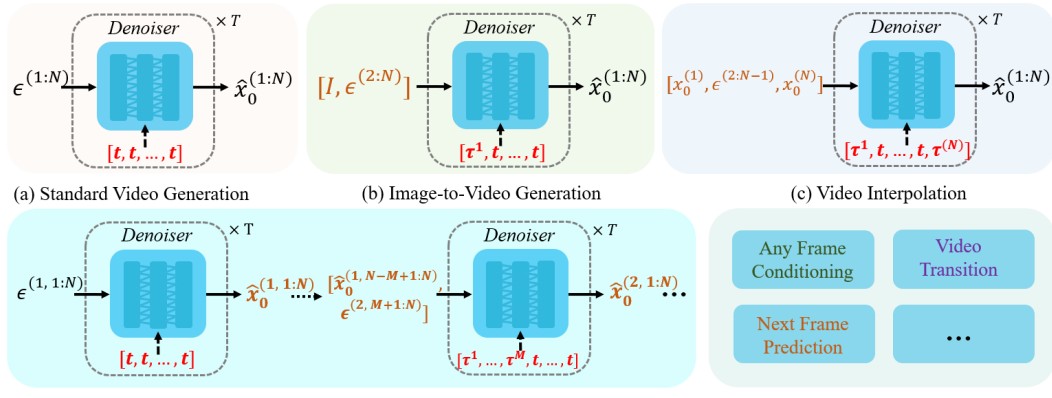

Figure 2: Diverse Applications of FVDM. (a) Standard Video Generation: Implements uniform timestep across frames, $[t, t, \ldots, t]$. (b) Image-to-Video Generation: Transforms a static image into a video using a customized vectorized timestep, $[\tau^1, t, \ldots, t]$, $\tau^1 \equiv 0$. (c) Video Interpolation: Smoothly interpolates frames between start and end, using $[\tau^1, t, \ldots, t, \tau^N]$, $\tau^1 = \tau^N \equiv 0$. (d) Long Video Generation: Extends sequences by conditioning on final frames, applying $[\tau^1, \ldots, \tau^M, t, \ldots, t]$, $\tau^1 = \ldots = \tau^M \equiv 0$ (e) Many More Zero-Shot Applications: Highlights potential for tasks such as any frame conditioning, video transition, and next frame prediction.

to vectorized timestep variable, we replace the original scalar timestep input, which had a shape of $(B)$, with a vectorized version $(B, N)$, where $B$ is the batch size and $N$ is the number of frames. Then, using sinusoidal positional encoding, we transform the input timesteps from shape $(B, N)$ to $(B, N, D)$, where $D$ is the embedding dimension. These vectorized timestep embeddings are then fed into the transformer block, where they condition both the attention and MLP layers through adaptive layer norm zero (adaLN-Zero) conditioning (Peebles & Xie, 2023). This process ensures that each frame's temporal dynamics are handled independently, resulting in improved temporal fidelity and noise prediction across frames.

**Training.** To address the potential computational explosion inherent in training diffusion models with vectorized timesteps, we introduce a novel *probabilistic timestep sampling strategy* (PTSS). In conventional VDMs, a scalar timestep $t$ is sampled for each batch element. However, when extending this approach to FVDM, where each frame evolve independently, the naive strategy of sampling a different timestep for each frame results in a combinatorial explosion, with $N$ frames yielding $1000^N$ combinations for 1000 timesteps, compared to just 1000 combinations for scalar timesteps. To mitigate this, we introduce a probability $p$ that governs the sampling process. With probability $p$, we sample distinct timesteps for each frame in the sequence, allowing for independent evolution. With probability $1 - p$, we sample the timestep for the first frame and let the other frames' timesteps be the same. This hybrid strategy significantly prevents excessive computational overhead and improves the standard video generation quality while retaining flexibility of frame-wise temporal evolution. An ablation study on different values of $p$ demonstrates the effectiveness of this approach, as shown in Fig. 3.

**Inference.** Despite using vectorized timesteps, our model remains compatible with standard diffusion sampling schedules like as DDPM (Ho et al., 2020) and DDIM (Song et al., 2020a). The PTSS allows the model to generalize effectively during inference, using established schedules without requiring new mechanisms. This balances the advantages of vectorized timesteps with the practicality of established diffusion model techniques, facilitating a smooth inference process.

## 2.4 APPLICATIONS

Beyond standard video generation, our Frame-aware Video Diffusion Model (FVDM) demonstrates remarkable versatility, performing a variety of tasks in a zero-shot manner, including image-to-video generation, video interpolation, and long video generation, as depicted in Fig. 2. The model's ability to flexibly manage complex temporal dynamics through the vectorized timestep variable $\tau(t)$ allows it to generalize to a broad range of video-related scenarios, extending well beyond conventional video synthesis.

**Standard Video Generation:** In the most basic application, the FVDM operates similarly to traditional video diffusion models. Every frame is initialized with $\epsilon^{(i)} = \mathcal{N}(0, \mathbf{I}), 1 \leq i \leq N$, the timestep is applied uniformly across all frames by setting $\boldsymbol{\tau}(t) = t \cdot \mathbf{1}$, where each frame evolves according to the same scalar timestep. This approach mirrors the dynamics of conventional diffusion models, where temporal coherence is maintained across frames.

**Image-to-Video Generation:** Our model is capable of generating dynamic video sequences from a static image $I$. By treating the image as the first frame, $\mathbf{x}_0^{(0)} = I$, we specially design $\tau^{(i)}(t)$, $1 \leq i \leq N$, for every frame. Experimentally, we find the simplest way to set the first frame noise-free $\tau^{(1)}(t) \equiv 0$, while set other frames with regular noise $\tau^{(i)}(t) = t, 2 \leq i \leq N$ yields satisfactory results. This formulation enables the smooth transformation of a still image into a coherent, multi-frame video sequence.

**Video Interpolation:** To interpolate intermediate frames between given starting and ending frames, similar to image-to-video generation, this intuitively way is to set the timesteps of the first and last frames to $\tau^{(1)}(t) = \tau^{(N)}(t) \equiv 0$, and applies regular noise to the intervening frames, i.e., $\tau^{(i)}(t) = t$ for $1 < i < N$. This indeed process results in the smooth synthesis of intermediate frames, ensuring seamless transitions between the start and end frames of the sequence.

**Long Video Generation:** Our model also supports the extension of video sequences by conditioning on the final frames of a previously generated clip. Similarly, given the last $M$ frames $\{\hat{\mathbf{x}}_0^{(k-1,i)}\}_{i=N-M+1}^N$ from the $(k-1)$th video clip, we generate the next video clip with $N-M$ new frames by setting $\tau^{(i)}(t) = 0$ for the first $M$ frames, where $\mathbf{x}_0^{(k,i)} = \hat{\mathbf{x}}_0^{(k-1,N-M+i)}$, and applying $\tau^{(i)}(t) = t$ for $M < i < N$. This method allows for seamless continuation of video sequences without temporal artifacts.

**Other Possible Applications:** Leveraging the flexibility of the VTV $\boldsymbol{\tau}(t)$, our FVDM has great potential to be extended to a multitude of tasks. For instance, videos can be generated from any arbitrary frame $\mathbf{x}_0^{(h)}, 1 \leq h \leq N$, by treating this frame as noise-free ($\tau^{(h)}(t) = 0$) and applying regular noise to the other frames ($\tau^{(i)}(t) = t$ for $i \neq h$). Additionally, by generating transitions between two videos, we can connect video clips or predict the next future frame similarly to long video generation but by generating only a single frame while maintaining the remaining frames from previous generations. Lastly, we think it should be very interesting to explore diverse inference schedules like noise progressively increase by frames, e.g., $\tau^{(i)}(t) = \inf(0.1 \cdot i \cdot t, t), 1 \leq i \leq N$ for image-to-video generation, and more complex applications like frame-level video editing (Meng et al., 2021) and video ControlNet (Zhang et al., 2023a) based on FVDM in the future.

## 3 EXPERIMENTS

### 3.1 SETUP

In this section, we detail the experimental setup for evaluating the proposed Frame-Aware Video Diffusion Model (FVDM). Our experiments are designed to assess the model's performance across a variety of tasks and compare it with state-of-the-art methods. We follow the principles of (Skorokhodov et al., 2022a) to evaluate our model with Fréchet Video Distance (FVD) (Unterthiner et al., 2019). Due to limited resources, we conducted ablation studies using a batch size of 3 for $200k$ iterations and trained our model for baseline comparison with a batch size of 4 for $250k$ iterations using two A6000 GPUs or one A800 GPU. We selected four diverse datasets for training and evaluation: FaceForensics (Rössler et al., 2018), SkyTimelapse (Xiong et al., 2018), UCF101 (Soomro, 2012), and Taichi-HD (Siarohin et al., 2019). We compared FVDM with several baselines for standard video generation, including MoCoGAN (Tulyakov et al., 2018), VideoGPT (Yan et al., 2021), MoCoGAN-HD (Tian et al., 2021), DIGAN (Yu et al., 2022), PVDM (Yu et al., 2023), and Latte (Ma et al., 2024).

### 3.2 ABLATION STUDY

We conducted a comprehensive ablation study to evaluate the impact of various hyperparameters and model configurations on standard video generation performance. All experiments were performed

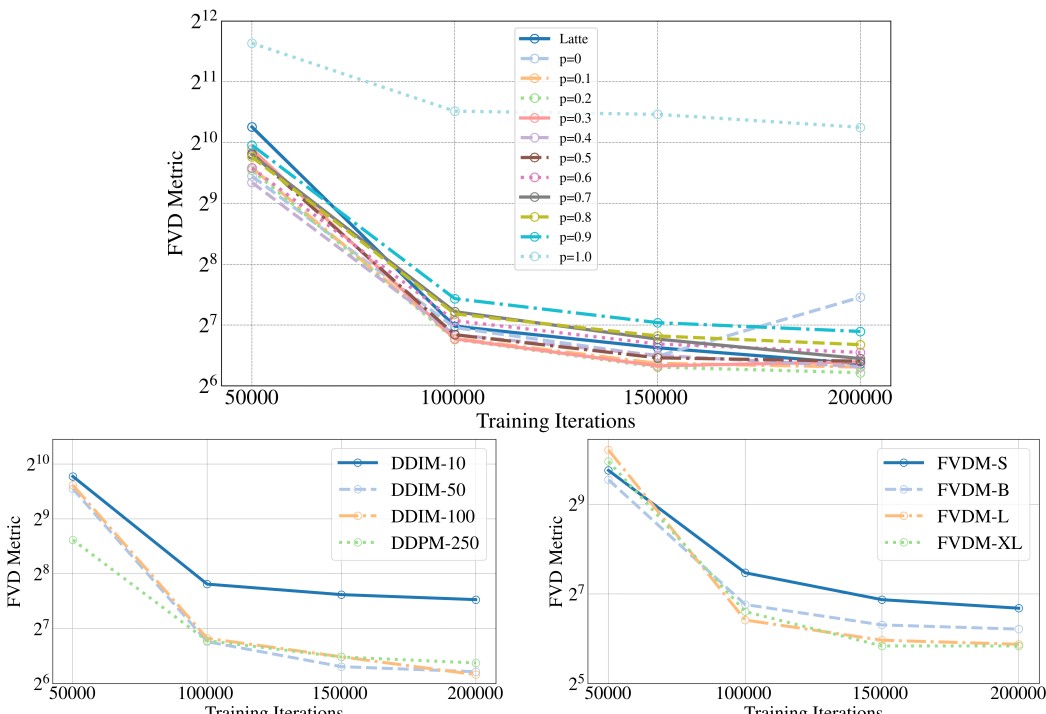

Figure 3: Comprehensive ablation study on FaceForensics dataset (Rössler et al., 2018) for video generation using FVD metric (lower is better) with training iterations from $50k$ to $200k$. Top, bottom left, and bottom right figures indicate ablation studies for sampling probability ($p$), inference schedule, and model scale, respectively.

| Method | FaceForensics | SkyTimelapse | UCF101 | Taichi-HD |
|---|---|---|---|---|
| MoCoGAN (Tulyakov et al., 2018) | 124.7 | 206.6 | 2886.9 | - |
| VideoGPT (Yan et al., 2021) | 185.9 | 222.7 | 2880.6 | - |
| MoCoGAN-HD (Tian et al., 2021) | 111.8 | 164.1 | 1729.6 | **128.1** |
| PVDM (Yu et al., 2023) | 355.92 | **75.48** | 1141.9 | 540.2 |
| Latte (Ma et al., 2024) | 77.70 | 110.45 | 604.64 | 267.12 |
| FVDM | **55.01** | 106.09 | **468.23** | 194.61 |

Table 1: FVD results comparing FVDM with the baseline on four different datasets. Lower FVD values indicate better performance. For Latte's result, we use the official code, and strictly follow the original configuration, except that we train it with batchsize 4 for $250k$ iterations and inference with DDIM-50, all the same as FVDM. Other results can be sourced in Ma et al. (2024); Skorokhodov et al. (2022b).

on the FaceForensics dataset (Rössler et al., 2018) and conducted with models of scale B, training with batch size 3, and inference with DDIM-50 (Song et al., 2020a) if no specification, using the FVD as the primary metric, where lower values indicate better performance. Fig. 3 presents the results of our ablation study graphically.

**Sampling Probability** The first part of our ablation study (Fig. 3) investigates the effect of the sampling probability $p$ in our PTSS. We observe that the model's performance is highly sensitive to this parameter, with $p = 0.2$ consistently yielding the best results across different training iterations. Notably, at $200k$ steps, $p = 0.2$ achieves an FVD score of 74.31, outperforming both the baseline Latte model (82.28) and other probability values. This finding suggests that a moderate level of probabilistic sampling strikes an optimal balance between exploration and exploitation during training.

**Sampling Schedule** In Fig. 3, we compare different sampling schedules, including DDPM (Ho et al., 2020) with 250 steps and DDIM (Song et al., 2020a) with varying step counts (100, 50, and 10). Our results indicate that DDPM-250 and DDIM with 100 or 50 steps perform comparably, with

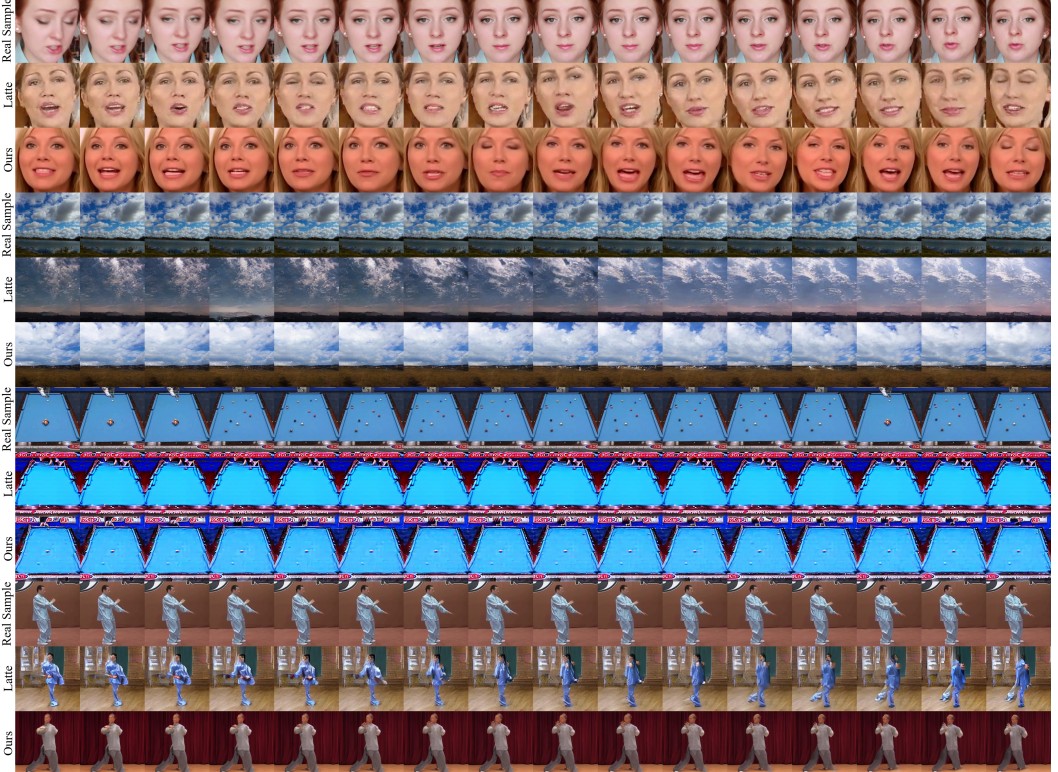

Figure 4: Qualitative comparison of real samples and generated video samples from FVDM/Ours and Latte (Ma et al., 2024) on four datasets, i.e., FaceForensics (Rössler et al., 2018), SkyTimelapse (Xiong et al., 2018), UCF101 (Soomro, 2012), and Taichi-HD (Siarohin et al., 2019) (from top to bottom). For a fair comparison, we select samples either of the same class w.r.t. UCF101 (Soomro, 2012) or with similar content w.r.t. other datasets. FVDM produces more coherent and realistic video sequences compared to the baseline.

DDIM-100 slightly edging out the others at $200k$ steps. However, DDIM-10 shows a significant performance degradation, suggesting that overly aggressive acceleration of the sampling process can be detrimental to generation quality. Based on these findings, we adopt the DDIM-50 schedule for our subsequent experiments, as it offers a good trade-off between efficiency and performance.

**Model Scale** The impact of model scale on generation quality is examined in Fig. 3. We evaluate four model sizes: *S* (32.59M parameters), *B* (129.76M parameters), *L* (457.09M parameters), and *XL* (674.00M parameters). Our results demonstrate a clear trend of improved performance with increasing model scale. The *XL* model consistently outperforms smaller variants, achieving the best FVD score of 57.25. This observation aligns with the scaling law (Kaplan et al., 2020).

### 3.3 STANDARD VIDEO GENERATION

In our evaluation of standard video generation, FVDM demonstrates superior performance compared to state-of-the-art methods. As shown in Table 1, FVDM achieves the lowest FVD scores on FaceForensics and UCF101, and the second lowest scores on other datasets, outperforming Latte and other leading models. This indicates enhanced video quality and temporal coherence.

FVDM leverages its innovative vectorized timestep variable to enhance temporal dependency modeling, which is evident in its ability to outperform Latte in most categories and maintain competitive performance in others. This effectiveness is further illustrated in Fig. 4, where qualitative comparisons reveal that FVDM generates video sequences with greater fidelity and smoother transitions compared to Latte. The visual results highlight FVDM's capacity to handle complex temporal dynamics, producing high-quality video outputs that closely mimic real-world sequences. This establishes FVDM as a robust and versatile tool in the realm of generative video modeling.

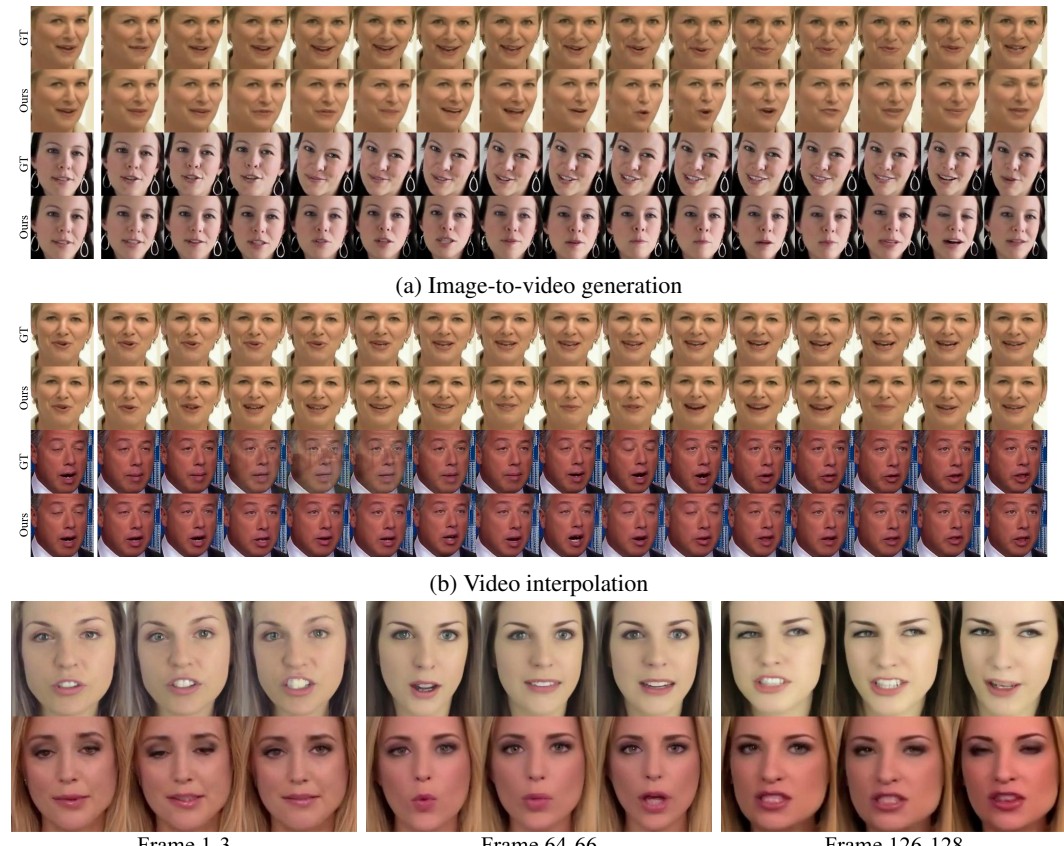

(a) Image-to-video generation

(b) Video interpolation

| Frame 1-3 | Frame 64-66 | Frame 126-128 |

(c) Long video generation

Figure 5: Zero-shot adaptations of FVDM on FaceForensics (Rössler et al., 2018). (a) image-to-video generation, where FVDM animates the first image into a coherent video sequence; (b) video interpolation, where FVDM generates smooth transitions between given the first frame and the last frame; and (c) long video generation, where FVDM generates long video sequences (we show 128 frames in this figure) while maintaining temporal coherence.

### 3.4 ZERO-SHOT APPLICATIONS OF FVDM

To demonstrate the versatility of FVDM, we evaluated its zero-shot performance on tasks such as image-to-video generation, video interpolation, and long video generation. Fig. 5 showcases qualitative results.

**Image-to-Video Generation:** As shown in Fig. 5(a), the model successfully generates a smooth and temporally coherent video from a single image, demonstrating its ability to infer motion and facial expressions without explicit training on such a task.

**Video Interpolation:** FVDM is also capable of generating smooth transitions between given start and end frames. Fig. 5(b) illustrates this capability, where the model interpolates between the first frame and last frame, creating a seamless video sequence that maintains the integrity of the original frames while filling in the intermediate motions.

**Long Video Generation:** One of the most challenging tasks for generative models is to produce long video sequences while maintaining temporal coherence. FVDM addresses this challenge by generating 128-frame videos that exhibit consistent motion and expression throughout the sequence, as depicted in Fig. 5(c). This demonstrates the model's ability to capture long-term dependencies in video data.

These zero-shot applications showcase the adaptability of FVDM across different video generation tasks, highlighting its potential for real-world applications where training data may be limited or diverse scenarios need to be addressed without prior fine-tuning. The model's performance in these

tasks is a testament to its robust architecture and the effectiveness of the vectorized timestep variable in capturing complex temporal dynamics.

## 4 RELATED WORK

The limitations in temporal modeling of conventional VDMs have led to a surge in approaches tailored to tasks. These methods predominantly rely on fine-tuning or employing zero-shot techniques to handle domain-specific challenges.

**Image-to-Video Generation.** Notably, DynamiCrafter (Xing et al., 2023a) introduces a model that animates open-domain images by utilizing video diffusion priors and projecting images into a context representation space. Furthermore, I2V-Adapter (Guo et al., 2023) presents a general adapter for VDMs that can convert static images into dynamic videos without altering the base model's structure or pretrained parameters. I2VGen-XL (Zhang et al., 2023b) addresses semantic accuracy and continuity through a cascaded diffusion model that initially produces low-resolution videos and then refines them for clarity and detail enhancement. Li et al. (2024) tackles fidelity loss in I2V generation by adding noise to the image latent and rectifying it during the denoising process, resulting in videos with improved detail preservation. Lastly, TI2V-Zero (Ni et al., 2024) introduces a zero-shot image conditioning method for text-to-video models, enabling frame-by-frame video synthesis from an input image without additional training or tuning.

**Video Interpolation.** MCVD (Voleti et al., 2022) stands out as the first to address this task using diffusion models, which presents a conditional score-based denoising diffusion model capable of handling future/past prediction, unconditional generation, and interpolation with a single model. Besides, LDMVFI (Danier et al., 2024) introduces a latent diffusion model that formulates video frame interpolation as a conditional generation problem, showing superior perceptual quality in interpolated videos, especially at high resolutions. Meanwhile, generative inbetweening (Wang et al., 2024b) adapts image-to-video models to perform high-quality keyframe interpolation, demonstrating the versatility of these models for video-related tasks. Finally, EasyControl (Wang et al., 2024a) transfers ControlNet (Zhang et al., 2023a) to video diffusion models, enabling controllable generation and interpolation with significant improvements in evaluation metrics.

**Long Video Generation.** On the one hand, ExVideo (Duan et al., 2024) enhances the video diffusion model's capacity to generate videos five times longer than the original model's duration through a parameter-efficient post-tuning strategy. Meanwhile, StreamingT2V (Henschel et al., 2024) introduces a conditional attention module and an appearance preservation module to generate long videos with smooth transitions through an autoregressive approach. Moreover, SEINE (Chen et al., 2023b) focuses on creating long videos with smooth transitions and varying lengths of shot-level videos through a random mask video diffusion model. On the other hand, FreeNoise (Qiu et al., 2023), FIFO-Diffusion (Kim et al., 2024), and FreeLong (Lu et al., 2024) achieve long video generation without additional training by noise rescheduling, iterative diagonal denoising, and SpectralBlend temporal attention, respectively.

## 5 CONCLUSIONS

We introduced the Frame-Aware Video Diffusion Model (FVDM), which addresses key limitations in existing video diffusion models by employing a vectorized timestep variable (VTV) for independent frame evolution. This approach significantly improves the video quality and flexibility of video generation across various tasks, including image-to-video, video interpolation, and long video synthesis. Extensive experiments demonstrated FVDM's superior performance over state-of-the-art models, highlighting its adaptability and robustness. By enabling finer temporal modeling, FVDM sets a new standard for video generation and offers a promising direction for future research in generative modeling. Potential extensions include better training schemes and different VTV configurations for other tasks like video infilling. In conclusion, FVDM paves the way for more sophisticated, temporally coherent generative models, with broad implications for video synthesis and multimedia processing.

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
