# OpenReview forum: "Redefining Temporal Modeling in Video Diffusion: The Vectorized Timestep Approach"
_ICLR.cc/2025/Conference — ICLR 2025 Conference Withdrawn Submission_

### Official Review · Reviewer_coPm · 2024-10-25

**Soundness:** 3
**Presentation:** 3
**Contribution:** 1
**Rating:** 1
**Confidence:** 5

**Summary:**

The paper proposes to modify the video diffusion framework. The main idea is to use different diffusion timestamps and different noise levels for different frames during training, this in turn allows for more applications during inference like image-to-video and long video generation.

**Strengths:**

- Supporting various applications with the same model seems very attractive and using different timestamps for different frames seems like a reasonable choice for it.

**Weaknesses:**

- Main idea of the paper is to model joint distribution P(x1, ... xn) with diffusion that applies different noise levels to x1, ..., xn. This idea was already proposed in [1], and in the context of video diffusion it was proposed in [2] and [3]. The difference between proposed approach and [2, 3] is only marginal, i.e. the way schedule for different timestamps is chosen.

[1] One Transformer Fits All Distributions in Multi-Modal Diffusion at Scale, Bao et al.
[2] Rolling Diffusion Models, Ruhe et al.
[3] Diffusion Forcing: Next-token Prediction Meets Full-Sequence Diffusion, Chen et al.

- Paper does not provide an explanation why FVDM performs better than Latte on pure video generation tasks. This seems counter intuitive (since FVDM was only training with same timestamps 80% of time) and may suggest some discrepancy in the evaluation setting.

- There is no comparison with autoregressive (masked) approaches on img-to-video and long video generation. Which questions the assumption that the proposed method has a good performance on these tasks.

**Questions:**

1) Why does the proposed method have better generation than the Latte baseline?
2) How does the proposed method compare to masking approaches, such as [4]?

[4] MAGVIT: Masked Generative Video Transformer, Yu et al.

---

### Official Review · Reviewer_kEDm · 2024-11-03

**Soundness:** 2
**Presentation:** 1
**Contribution:** 1
**Rating:** 5
**Confidence:** 4

**Summary:**

The work proposes and analyzes a noise schedule for video diffusion models that uses non-shared noise strengths per each frame in a video. This modification unlocks applications in a zero-shot manner:

**Strengths:**

- The new schedule unlocks many zero-shot capabilities, like image-to-video, video interpolation and video extrapolation
- The proposed modification is extremely simple and hence could be easily adopted by many others.
- While it's something intuitive, the authors still showed that the diffusion framework does not break by doing the necessary derivations.
- The writing is clear

**Weaknesses:**

- While in the description, the idea looks concise and neat, the model had to be trained with the extra complication of using the vanilla strategy of sharing timesteps for most frames (setting p<=0.3 seems to yield the optimal results) — otherwise there would be too much quality loss. In this way, I feel like the method does not quite work.
- The paper provides no qualitative video samples at all and expects a reader to assess the visual quality by inspecting static grids of frames, which makes it impossible to confidently compare with the baselines or detect any particular artifacts (if they exist).
- I believe it would be more fair to compare against the standard diffusion schedule for the analyzed downstream tasks: with and without reconstruction guidance (like in Ho et al "Video Diffusion Models")
- The qualitative results in Figure 4 look worse for some classes (e.g. billiard) look worse than for the contemporary methods. It would be fair to include more recent methods in ablations.

**Questions:**

- Please, provide the qualitatives
- Why there is such a huge quality loss for p=1? Doesn't it mean that the diffusion embedding is broken?
- Is it possible to use a simple strategy for timestep conditioning? (e.g., conditioning on a concatenated timestep embedding from all the frames)

---

### Official Review · Reviewer_G5Fa · 2024-11-04

**Soundness:** 3
**Presentation:** 2
**Contribution:** 3
**Rating:** 3
**Confidence:** 3

**Summary:**

This paper introduces the Frame-Aware Video Diffusion Model (FVDM), a novel approach to video generation that leverages a vectorized timestep variable (VTV) to overcome the limitations of traditional video diffusion models (VDMs). Existing VDMs apply a scalar timestep uniformly across all video frames, which constrains their capacity to model complex temporal dependencies. FVDM addresses these issues by allowing each frame to evolve independently, resulting in enhanced temporal fidelity. The model demonstrates strong performance in multiple video generation tasks, including standard video synthesis, image-to-video generation, video interpolation, and long video generation, all performed in a zero-shot manner without retraining.

**Strengths:**

1. The motivation for the proposed FVDM is fundamental and addresses a clear limitation in existing video diffusion approaches. The introduction effectively communicates the need for a more flexible temporal modeling framework.
2. The use of a vectorized timestep variable (VTV) to enable independent frame evolution represents a significant improvement over conventional methods, allowing the model to capture fine-grained temporal dependencies more effectively.
3. The proposed FVDM is highly versatile, supporting multiple video-related tasks without requiring retraining, which is beneficial for practical deployment scenarios where fine-tuning for each task is not feasible.
4. From the visualizations, it is evident that the model maintains better consistency in facial details and actions, demonstrating improved fidelity and coherence compared to baseline models.

**Weaknesses:**

1.The method proposed in this paper seems to perform well in face-related scenes, as seen in Figure 4, but for natural landscapes or sports activities, the frame-to-frame variation is not as significant, which may limit its applicability to a wider range of scenarios. As shown in Table 1, the method performs well on the FaceForensics and UCF101 datasets, but it does not achieve optimal performance on the SkyTimelapse and Taichi-HD datasets.
2. The paper lacks pseudocode for the VTV method, which makes it challenging to reproduce the reported results. Additionally, the experimental setup in Section 3.1 provides insufficient training details, such as information about the learning rate, optimizer settings, and other hyperparameters.
3. The impact of using the vectorized timestep variable (VTV) on inference efficiency is not discussed in the paper. It would be helpful to understand whether the use of VTV affects inference speed and, if so, to what extent.

**Questions:**

1. How does the inference time of the proposed model compare to other baseline models for long video generation?
2. Are there any strategies for the VTV method to adjust the temporal coherence at different levels or make the model focus more on specific aspects of the video sequence?

---

### Note · Authors · 2024-11-14

I have read and agree with the venue's withdrawal policy on behalf of myself and my co-authors.